# Impact of Technology-Based Intervention for Improving Self-Management Behaviors in Black Adults with Poor Cardiovascular Health: A Randomized Control Trial

**DOI:** 10.3390/ijerph18073660

**Published:** 2021-04-01

**Authors:** Tulani Washington-Plaskett, Muhammed Y. Idris, Mohamed Mubasher, Yi-An Ko, Shabatun Jamila Islam, Sandra Dunbar, Herman Taylor, Arshed Ali Quyyumi, Priscilla Pemu

**Affiliations:** 1Cardiovascular Medicine, Boston University School of Medicine, Boston, MA 02118, USA; Tulani.Washington-plaskett@bmc.org; 2Department of Medicine, Morehouse School of Medicine, Atlanta, GA 30310, USA; myidris@msm.edu (M.Y.I.); htaylor@msm.edu (H.T.); 3Community Health and Preventive Medicine, Morehouse School of Medicine, Atlanta, GA 30310, USA; mmubasher@msm.edu; 4Department of Biostatistics and Bioinformatics, Rollings School of Public Health, Emory University, Atlanta, GA 30322, USA; yi-an.ko@emory.edu; 5Emory Clinical Cardiovascular Research Institute, Emory University, Atlanta, GA 30322, USA; shabatun.jamila.islam@emory.edu (S.J.I.); aquyyum@emory.edu (A.A.Q.); 6Nell Hodgson Woodruff School of Nursing, Emory University, Atlanta, GA 30322, USA; sbdunba@emory.edu; 7Cardiovascular Research Institute, Morehouse School of Medicine, Atlanta, GA 30310, USA; 8Department of Medicine, Emory University School of Medicine, Atlanta, GA 30322, USA

**Keywords:** equity, health disparities, behavioral intervention technology, Life’s Simple 7, CVD, Blacks

## Abstract

Cardiovascular disease (CVD) is the number one killer of adults in the U.S., with marked ethnic/racial disparities in prevalence, risk factors, associated health behaviors, and death rates. In this study, we recruited and randomized Blacks with poor cardiovascular health in the Atlanta Metro area to receive an intervention comparing two approaches to engagement with a behavioral intervention technology for CVD. Generalized Linear Mixed Models results from a 6-month intervention indicate that 53% of all participants experienced a statistical improvement in Life’s Simple 7 (LS7), 54% in BMI, 61% in blood glucose, and 53% in systolic blood pressure. Females demonstrated a statistically significant improvement in BMI and diastolic blood pressure and a reduction in self-reported physical activity. We found no significant differences in changes in LS7 or their constituent parts but found strong evidence that health coaches can help improve overall LS7 in participants living in at-risk neighborhoods. In terms of clinical significance, our result indicates that improvements in LS7 correspond to a 7% lifetime reduction of incident CVD. Our findings suggest that technology-enabled self-management can be effective for managing selected CVD risk factors among Blacks.

## 1. Introduction

Cardiovascular disease (CVD) is the number one killer of men and women in the U.S., with marked ethnic/racial disparities in prevalence, risk factors, associated health behaviors, and death rates [1,2]. There are a multitude of factors that may predispose Blacks to increased CVD risk. Individual-level characteristics, including socio-economic status, education level, and health literacy contribute to the CVD disparities observed between Blacks and Whites and within the Black community [1,3,4]. So too does a higher prevalence of hypertension, diabetes, and obesity in Black communities, which is associated with ethnic disparities in health behaviors that contribute to CVD risk [1,5,6]. These factors include higher levels of physical inactivity, smoking, greater than moderate drinking, dietary factors, and psychological distress which are more common among Blacks compared to Whites [1,6,7]. Healthy behaviors as well as psychosocial well-being are modulated by neighborhood factors (e.g., availability of food choices, walkability, social cohesion, violence) [8,9,10,11]. In this way, environmental factors are also key to understanding differences in cardiovascular health with some geographies having higher than expected cardiovascular mortality, hospitalizations, and emergency department visits (at-risk) than others (resilient) [12].

The American Heart Association has defined ideal cardiovascular health based on the management of seven risk factors (Life’s Simple 7, LS7): smoking status, physical activity, weight, diet, blood glucose, cholesterol, and blood pressure [13]. Previous studies have demonstrated that behavioral interventions can contribute to the cessation of smoking [14,15], increased physical activity [16,17,18], enhanced nutrition [19,20], and reduced blood sugar, cholesterol, and blood pressure levels [20]. Patients with ideal health in five or more LS7 risk factors over a 4–7-year period enjoyed close to an 80% reduction in mortality and an almost 90% drop in mortality associated with CVD [21]. The efficacy of self-management interventions to improve health behaviors and address CVD risk factors for low income and minority populations can also be further enhanced by social support factors, including family and peer support [4,15,16]. In sum, while changing behavior can be difficult, self-management is critical for preventing and managing CVD. However, effective solutions like targeting and tailoring of lifestyle interventions are labor intensive and can be cost prohibitive.

Behavioral intervention technologies can provide scalable solutions to support patients in actively managing their health. Behavioral intervention technologies allow for the tailoring of tools to the individual, timely information delivery, standardized messaging, layered content for more motivated participants, and the potential for greater efficiency/cost savings. With the rise of e-health, behavioral intervention technologies have been shown to provide effective and sustainable self-management tools for patients with chronic diseases. Yet, these benefits have not extended to minority and other underserved patient populations and in turn may exacerbate health disparities [22]. The barriers to minority populations’ engagement with e-health solutions and interventions include lack of perceived value, such technologies creating more work, limited health and technology literacy, cognitive and physical disabilities, lack of cultural relevance, limited access to computers/hardware, privacy/trust concerns, technical problems, and unclear or confusing instructions on the use of e-health technologies [23].

The purpose of this study was to drive health behavior change among Blacks with poor cardiovascular health in the Atlanta Metro area, while addressing barriers to the effective use of behavioral intervention technologies for self-management of CVD. Specifically, we wanted to test whether a health coach trained to reinforce behavior change in addition to Health360x, a behavioral intervention technology developed at Morehouse School of Medicine for self-management of chronic conditions, improves cardiovascular health profiles (as defined by LS7) when compared to the use of Health360x alone. In previous work, we showed that coaching can help maintain engagement in technology-enabled behavioral interventions and improve self-efficacy for behavior change through a combination of goal setting, self-monitoring, and problem solving among other evidence-based behavior change strategies [24]. Our hypothesis is that an intervention combining Health360x with a health coach (high tech-high touch) will be more effective in improving cardiovascular health of patients with low LS7 scores as we believe it will help with patient engagement and attainment of self-management goals. 

## 2. Methods

### 2.1. Participants and Recruitment

In order to investigate the impact of a technology-based intervention on behavior change among Blacks in Atlanta who have high cardiovascular risk, we recruited participants through the Morehouse-Emory Cardiovascular (MECA) Center for Health Equity study. The goal of the overall parent study was to explore individual and neighborhood factors that are protective against poor cardiovascular health among Blacks in greater Atlanta [25]. The Metro Atlanta is ideal for assessing the impact of technology-based intervention on improvements in CVD risk among Blacks for two reasons. First, there are several CVD risk factors in Georgia which exceed the national average: the prevalence of hypertension, diabetes, and physical inactivity are 30.4%, 10.1%, and 24.7% compared to the national prevalence of 27.8%, 8.0%, and 23.0%, respectively [26,27]. Second, there is rich diversity in ethnic origin and socioeconomic status within the Black community in Atlanta. 

Participants from the baseline MECA clinical study completed patient visits at the Emory University Hospital, Emory Clinical Cardiovascular Research Institute (ECCRI), and Morehouse School of Medicine in order to measure risk factors and calculate Life’s Simple 7 (LS7) scores. The dietary intake of fruits and vegetables was assessed using the Block Fruit and Vegetable Screener [28] and the Delta NIRI Food Frequency Questionnaire for African Americans [29] and physical activity using the Paffenbarger Physical Activity Questionnaire [30]. Blood pressure readings were conducted three times with the volunteer at rest in the sitting position and categorized according to the Joint National Committee on BP recommendations [31]. To calculate Body Mass Index (BMI), weight (kg) was measured with the participant wearing street clothes without shoes and height (m) was assessed with the participant standing on a flat surface against a wall. Waist and hip circumferences were obtained using a non-elastic tape measure midway between the lowest rib margin and the iliac crest, 1 inch above the umbilicus following established guidelines. Blood specimens (120 mL) were also collected after overnight fasting for testing blood glucose/insulin and a full lipid panel (total, low density, high density cholesterol, and triglyceride levels) as well as an extended chemistry panel and complete blood count using standard methods. LDL-C was determined using Friedwald’s equation using results from lab testing. 

The results from this data collection were used to calculate LS7 using a scoring algorithm adopted from the Goals and Metrics Committee of the Strategic Planning Task Force of the American Heart Association [32].
BMI: Poor = ≥ 30.0 kg/m^2^; Intermediate = 25.0–29.9 kg/m^2^; Ideal = <25.0 kg/m^2^;Cholesterol: Poor = ≥ 240 mg/dL; Intermediate = 200–239 mg/dL (untreated) or treated to goal; Ideal = < 200 mg/dL (untreated);BP: Poor = SBP ≥ 140 mmHg or DBP ≥ 90 mmHg; Intermediate = Systolic Blood Pressure (SBP) 120–139 mmHg, Diastolic Blood Pressure (DBP) 80–89 mmHg, or treated to goal; Ideal = SBP < 140 mmHg and DBP < 90 mmHg;Fasting glucose: Poor = ≥ 126 mg/dL; Intermediate = 100–125 mg/dL (untreated) or treated to goal; Ideal = <100 mg/dL (untreated).

In addition, participants who were currently smoking were classified as Poor, those who were former smokers, but quit ≤ 12 months ago were classified as Intermediate, and those who had never smoked or quit >12 months ago fell into the Ideal LS7 category. Finally, a healthy diet score was assessed based on the 5 aspects of diet selected by the American Heart Association: fruits/vegetables (≥ 4.5 cups/day), fish (≥two 3.5-oz servings/week), fiber-rich whole grains (≥ 1.1 g of fiber per 10 g of carbohydrates, and sodium. Participants with 0 or 1 components were classified as Poor, 2–3 components Intermediate, and 4–5 components as Ideal. The LS 7 score ranges between 0–14, giving 2 points for Ideal, 1 point for Intermediate, and 0 points for Poor components [33].

All participants in the baseline clinical study with low LS7 scores who agreed to be contacted for this study were contacted by the research staff. To reach recruitment goals, participants were also recruited from databases where volunteers consented to be contacted for research studies (including ResearchMatch.org, accessed on 28 July 2021), through self-referral in response to advertising placed with community partners, as well as radio and television ads. At the time of recruitment, the following contact information was collected from participants: (i) a telephone number capable of receiving text messages, (ii) email address, (iii) home address, and (iv) email and street addresses of two personal contacts of the participant. An enrollment package was sent by email to the interested participants, including introductory study information and an online screening form. Non-responders received follow-up calls from study staff and live chat sessions were organized to answer questions about the project. Once participants completed the study forms, enrollment information, and directions on how to complete informed consent online were provided to those who met study inclusion criteria. Once the consent process was completed, training on the use of Health360x began. All protocols were approved by Emory University and Morehouse School of Medicine IRBs.

#### Inclusion and Exclusion Criteria

In order to be eligible for the study, participants needed an LS7 composite score of 8 or lower, access to the internet, a self-reported ability to participate in increased physical activity, and English fluency. The exclusion criteria included history of coronary artery disease (CAD) documented by CAD diagnosis or prior acute myocardial infarction, percutaneous coronary intervention, coronary artery bypass surgery, or chronic angina; aortic stenosis; history of chronic diseases that may alter brachial artery flow-mediated vasodilation measurements, such as peripheral vascular disease, HIV/AIDs, lupus, and cancer. Additionally, we excluded participants unwilling to use the internet and those with a history of alcohol or drug abuse or psychiatric diagnosis that would interfere with their ability to participate. Pregnant and/or breastfeeding women were also excluded as were participants with cognitive deficits severe enough to preclude meaningful participation.

### 2.2. H360x Intervention for Cardiovascular Disease Self-Management

This study provides a theory-based approach to engaging a vulnerable population in a technology-enabled behavioral intervention. The intervention is designed to compare two approaches to engagement (technology alone versus technology coupled with a health coach) and behavior change. The underlying construct of behavioral change (with Health360x) is a system that frames behavior as changeable and adaptable in a bidirectional manner, based on capability, opportunity, and motivation. Capability is the psychological and physical capacity to engage in the activity concerned, including the necessary knowledge and skills. Motivation is largely governed by the cognitive processes that energize and direct behavior, including goal-directed conscious decision-making, habitual processes, emotional response, as well as analytical decision-making. Opportunity refers to factors that lie outside the individual, that facilitate or prompt the behavior. Enhanced self-efficacy for behavior change in turn leads to improved cardiovascular health outcomes.

Health360x (Accuhealth, Atlanta, USA) is a patented behavioral intervention technology platform developed at the Morehouse School of Medicine to assist with chronic illness care [24,34]. It is available as a web-based or mobile application and supports behavior change by providing functionality for improving health literacy and self-efficacy through built-in coaching support for accountability and problem solving. The application also provides a social networking forum to promote motivation and community, as well as a curriculum and health tracker that can record blood pressure, BMI, physical activity, and self-management goals. The selected, consenting participants were trained on (a) how to create a user profile with brief demographic information, (b) how to participate in a forum, (c) how to search and browse, and (d) how to send private messages and post comments. All participants were trained to use the technology-based smart application to monitor the management of their CVD risk factors including blood sugar, hypertension, BMI, and total cholesterol, while improving their lifestyles in terms of diet, smoking, and physical activity.

After completing training, participants completed a brief test of their ability to navigate and use Health360x. Specifically, we tested a user’s ability to sign on, navigate to different parts of the application, upload and view data, set and track behavioral goals, find and complete study-related questionnaires, send emails to the study team, post to an online community forum, and access educational content. Once a participant successfully completed the test, they were randomized to the respective clinical intervention arm: Arm A (high tech only) or Arm B (high tech-high touch). Coaches met with participants in Arm B in person or by phone every week for the 1st month, then every two 2 weeks for 8 weeks, and finally monthly for 3 months. The purpose of each visit was to advance a patient’s self-efficacy for self-management behaviors. The coaches were trained to use appreciative inquiry techniques to identify barriers to behavior change and co-develop a plan to address barriers with their participants. All participants were provided 24/7 technical support, but participants in Arm B had access to coaches and the ability to ask questions using secure messaging within Health360x, including technical assistance.

#### Randomization

The underlying implementation study design was based on a parallel controlled permuted block randomization scheme. We used variable-block size randomization algorithm to eliminate selection bias and to ensure intermittent as well as overall balance in outcome predictor variables between those assigned to Arm A (Health360x alone) or Arm B (Health360x and health coach). The participants were stratified based on sex-at-birth and neighborhood-community (at-risk, resilient). Census tracts with higher (lower) than expected cardiovascular deaths, emergency department (ED) visits, and hospitalizations for black adults aged 35 to 64 from 2010 through 2014 were classified as at risk (resilient) [12]. The high tech-high touch group (Arm B) received Health360x and a health coach who helped them create personalized action plans (in Health360x). Figure 1 provides a flowchart including counts of participants who were screened, randomized. 

### 2.3. Statistical Methods

#### 2.3.1. Outcomes, Mediators and Effect-Modifier Variables

The primary outcome variables considered here are the 6 month-change in LS7 (e.g., smoking status, physical activity, weight, diet, blood glucose, cholesterol, and blood pressure). These health behaviors and metrics represent seven out of the top 10 most costly CVD risk factors [35]. The LS 7 score ranges between 0 and 14, giving 2 points for ideal, 1 point for intermediate, and 0 points for poor components [33]. Additional outcome measures were compiled from the 6-month changes in the underlying CVD risk variables. This included improvements in LS7 (Δ > 0), blood pressure (Δ mmHg < 0 in both SBP and DBP), Blood glucose (Δ < 0) and BMI (Δ < 0)

The baseline covariates employed in these analyses included age, sex at birth, highest level of education, and neighborhood classification (at risk, resilient, or neither) as well as the baseline measurements of the clinical variables and LS7. Multivariable analyses were further adjusted for an engagement variable to measure the use of Health360x. The engagement variable for using Health360x was captured through the following computed variables: number of successful logins, average interval between successful logins, median interval between successful logins, number of sessions as defined by app usage occurring after >= 5 min of no activity, average session duration, median session duration, average lapsed time between sessions, and median lapsed time between sessions.

#### 2.3.2. Sample Size Calculation

The original power calculations of this study used preliminary data of the reported LS7 score [36] with a two-sided 0.05 significance level, 20% attrition rate, and a sample size of 120 (*n* = 60 per group), yielding 80% power to detect a 1.04 difference in the overall LS7 score at 6 months between the two randomization arms.

#### 2.3.3. Statistical Analysis

The participants’ characteristics, including demographics, baseline, and 6-month changes in CVD risk variables were summarized and compared by the randomization groups (coaching vs. no coaching). Binary/categorical variables like sex at birth, education, and the presence of diabetes were summarized using frequencies and percentages. Continuous variables like age, BMI, and blood glucose were summarized by means and standard deviations. Baseline characteristics and outcome variables of CVD risk factors were compared between randomization groups using Chi-Square/Fisher exact tests for categorical variables and the t-test/Mann–Whitney test for continuous variables. For the evaluation of baseline balance, we also utilized a univariate logistics function with the likelihood of being on one of the randomization groups as an outcome vis-à-vis each of the potential predictors of the outcome measures, to determine any association with patient’s baseline characteristics Statistically insignificant p-values indicate that the patient characteristic (i.e., age) is not correlated with assignment to randomization group suggesting comparability between groups. 

Generalized Linear Mixed Models (GLMM) techniques with several iterations were pursued for multivariable analyses due to their capabilities to accommodate (1) infusing a random effect component (census tract/neighborhood), (2) affording participants their own random intercept and (3) normally, non-normally- distributed, and binary outcomes using the identity or the logit link. The effect of the intervention (Health360x and a health coach) on study outcomes was examined using GLMM to adjust for demographics, potential confounding variables, and effect modifiers variables, as well as frequency and duration of use of the Health360x. The overall α-level was set at 0.05 while adjusting for multiple comparisons, if needed. SAS version 9.4 (SAS Institute, Cary, USA) was used for analyses.

## 3. Results

A total of 120 eligible participants (*n* = 58 vs. *n* = 62 in the coaching vs. no coaching groups) were included in these analyses. Table 1 provides an overview of the participants’ characteristics by study arm. In Table 2, we provide participants details on LS7 and its components at baseline and at 6 months to characterize CVD risk burden by study arm. To emphasize engagement with Health360x as well as CVD risk burden using visual description, we provide graphical representations of number of successful logins, LS7, diastolic and systolic BP, BMI and smoking status by randomization group in Figure 2, Figure 3, Figure 4, Figure 5, Figure 6 and Figure 7. 

The randomized groups (coaching vs. no coaching) indicated 51.7% vs. 54.8% improvement in 6-month LS7, *p*-value = 0.73. Figure 8 provides a cumulative distribution function of 6-month change in LS7 between randomization groups. In addition, we found improvements by randomized groups 43.1% (coaching) vs. 29.0% (no coaching) in blood pressure (improvement in both SBP and DBP), *p*-value = 0.11; 51.7% (coaching) vs. 56.4% (no coaching) in BMI, *p*-value = 0.6038 and 63.8% (coaching) vs. 58.1% (no coaching) in Blood glucose, *p*-value = 0.5209. Additionally, smoking status comparison between baseline and 12 months did not change significantly.

The main effects GLMM indicated no statistically significant differences between the coaching and no coaching groups in changes (as continuous variables) in LS7, SBP, DBP, BMI, or time spent in physical activities (Table 1 and Table 3). However, females relative to males and independent of their randomization group assignment, demonstrated a statistically significant improvement in the 6-month change in BMI and DBP (an average improvement, respectively, of 1.16; *p*-value = 0.01 and 4.1 mmHg, *p*-value = 0.04). Additionally, females (versus males) also indicated a significant reduction in time spent in physical activities (see Table 3a–c).

Multiple comparisons-adjusted subgroup analyses using GLMM interaction models indicated that the coaching group (vs. the no coaching group) had a statistically significant advantage in improvement in LS7 for those who resided in the “at risk” neighborhoods (an average improvement of 1.13; *p*-value = 0.04). Additional main and interaction (randomization group with neighborhood type/gender) GLMM analyses using improvement (as a yes/no binary outcome) in LS7, blood pressure, blood glucose, and BMI indicated no statistically significant advantage for the coaching vs. the no coaching (see Table 4).

## 4. Discussion

### 4.1. Principal Findings

The study adopted a stratified permuted block randomization scheme to balance all baseline characteristics, including CVD risk factors. Though statistical significance between coaching and no coaching groups was not reached, results indicated that 53% of all participants using Health360x experienced improvement (of Δ ≥ 1) in LS7, 54% in BMI, 61% in blood glucose, and 53% in SBP, indicating that technology-enabled self-management can be effective for managing selected CVD risk factors among Blacks. It is worth noting that those who demonstrated improvement in their LS7 at 6 months (*n* = 64, mean Δ = 1.7) had a median reduction of 6mmHg in SBP, 8mg/dl in BG, 0.15 units in BMI, and a 30-minute increase per week in time spent in physical activity. It is also imperative to point out that the overall mean increase of 0.5 in LS7 at 6 months suggests an average reduction of 6.5% in the risk of CVD [37]. In our study and irrespective of the intervention effect, females at 6 months demonstrated a statistically significant improvement in BMI and diastolic blood pressure. Interestingly, however, females also showed significant reduction in self-reported time spent in physical activities, suggesting presence of individual differences in how self-management, like dieting and self-efficacy, can influence health. Finally, though the results suggest mixed evidence of health coaching (above and beyond technology) influencing cardiovascular health, we found no significant differences between randomized groups in changes expressed in continuous or binary (yes/no) units of LS7 composite scores or their constituent parts between baseline and 6 months. However, additional subgroup analyses indicated strong evidence that health coaches can help improve overall LS7 in at-risk neighborhoods.

### 4.2. Limitations

We identified a few limitations of our study. First, the technology infrastructure underlying Health360x went through some significant changes during the intervention. While the features and functionality of the platform remained the same, the backend was revamped which led to participants facing some technical issues during migration (e.g., unable to sign-in to profiles) and differences in how usage data was defined and collected. The latter issue around usage data limited our measurement of user engagement with Health360x to variables that captured when and how long participants used Health360x. We were unable to capture “how” (i.e., what features/functionality) participants were using the platform which was a second limitation of our study. There are many ways in which a participant can use Health360x after logging on (i.e., tracking health, setting goals, etc.). Understanding how participants are using the application and the factors that lead them to oscillate between usage and inactivity are essential to understanding the minimum necessary “dose” of engagement to gain benefit.

Another possible explanation for the inability to reach statistical significance between coaching and no coaching group is the presence of a relatively higher heterogeneity in the outcome measures that may have diminished the effect size. This issue could be addressed by having a larger sample size and/or longer follow up. 

A final limitation comes from the recurring realization in the recruitment of coaches and implementation of the study: not all coaches are created equal. Coaches varied in terms of their motivation for participation in the study, their professional background, and lived experiences all of which influences interactions with participants. The coaches were trained on Health360x and completed modules on appreciative inquiry but would have benefited from more uniform training in cultural congruence and cognitive behavioral therapy or motivational interviewing.

## 5. Conclusions

In this study, we determined the impact of a technology-based intervention (Health360x) for improving self-management of Morehouse-Emory Cardiovascular (MECA) recruited participants in terms of CVD risk factors. The overall goal of MECA is to investigate the social and environmental factors that lead to risk and resilience among Blacks, with a geographic focus on the Atlanta Metro area. In this clinical intervention study, we explored the role of coaching on improving CVD risk factors and found evidence suggesting that health coaching can help promote engagement in and adherence to technology-enabled self-management interventions and in turn cardiovascular health. The findings also suggest the need for longer follow-on studies with a larger sample size, featured by closer monitoring of engagement and use of technology-based behavioral health intervention technologies like Health360x.

## Figures and Tables

**Figure 1 ijerph-18-03660-f001:**
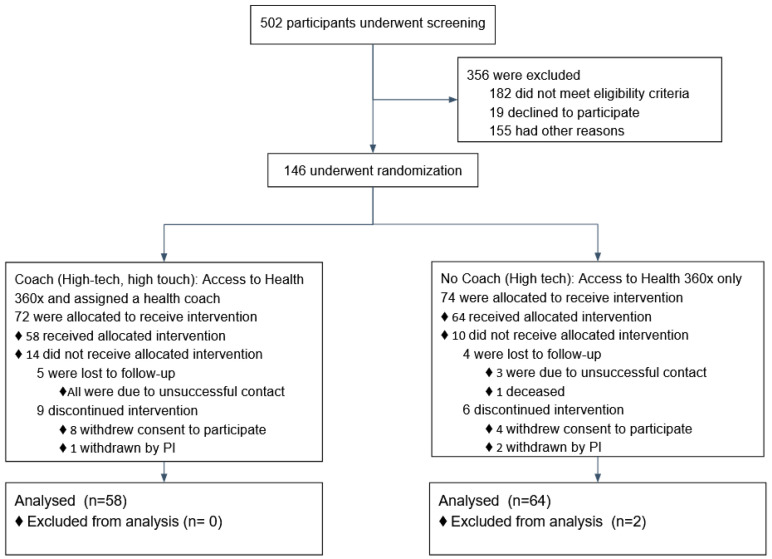
Flowchart of screening, randomization, and follow up.

**Figure 2 ijerph-18-03660-f002:**
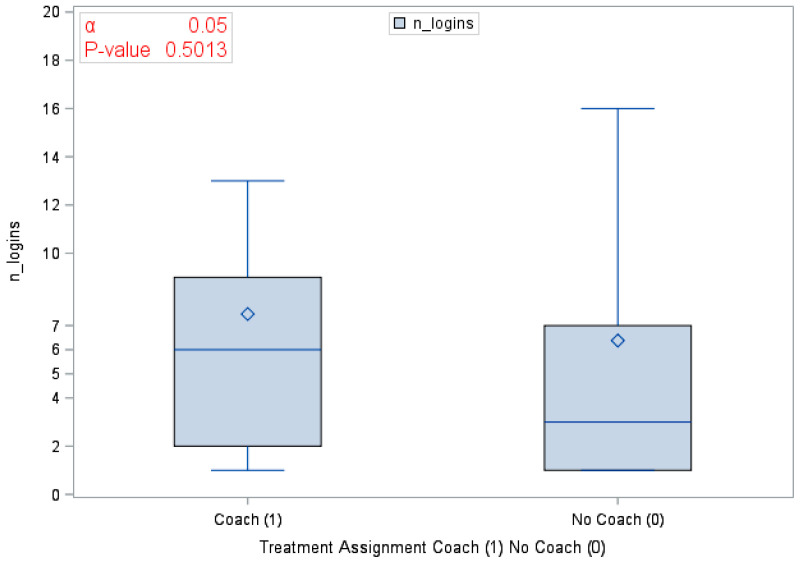
Number of successful logins by randomization group.

**Figure 3 ijerph-18-03660-f003:**
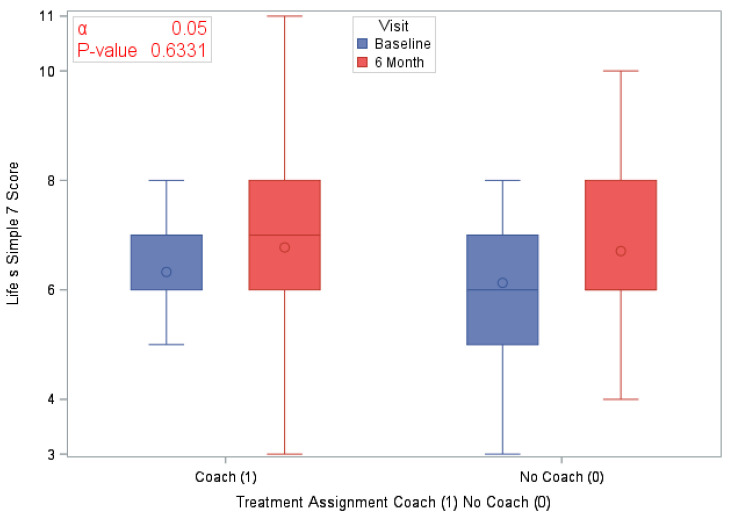
Life’s Simple 7 by study visit and randomization group.

**Figure 4 ijerph-18-03660-f004:**
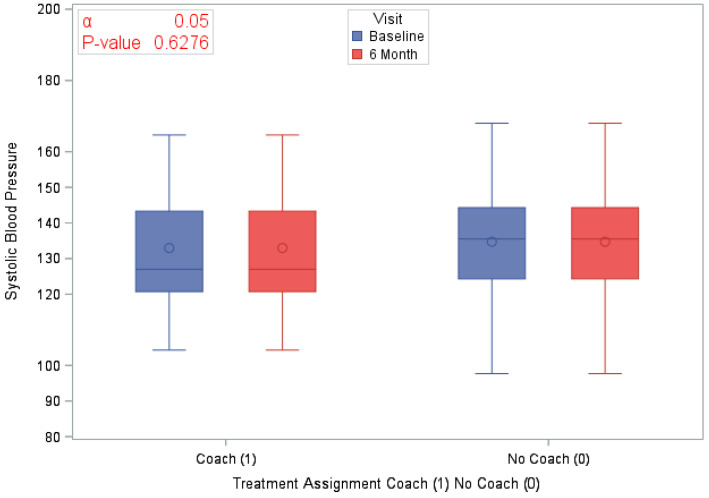
Systolic blood pressure by study visit and randomization group.

**Figure 5 ijerph-18-03660-f005:**
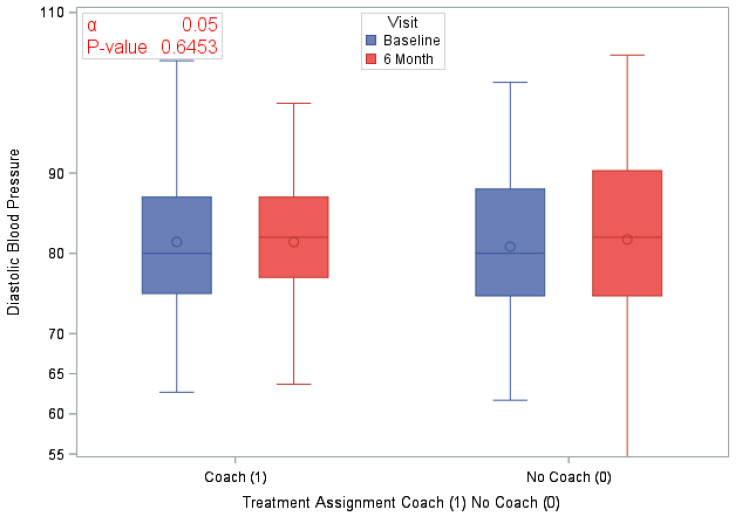
Diastolic blood pressure by study visit and randomization group.

**Figure 6 ijerph-18-03660-f006:**
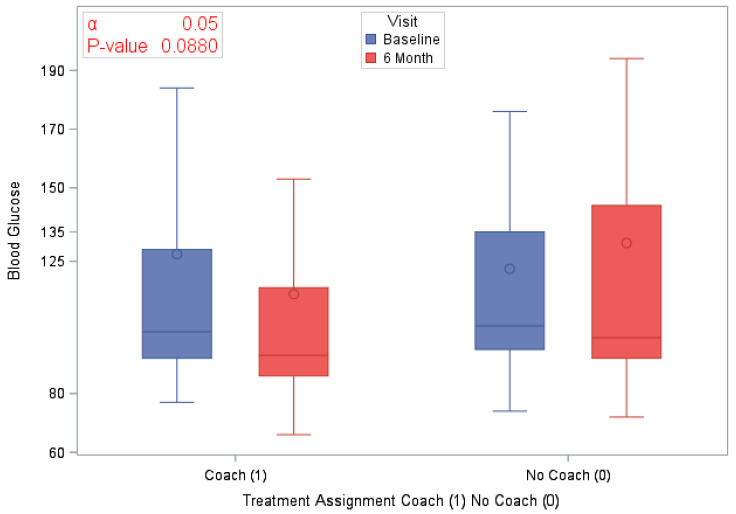
BMI by study visit and randomization group.

**Figure 7 ijerph-18-03660-f007:**
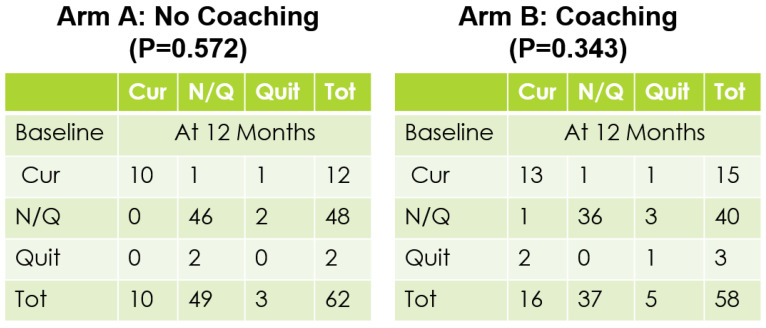
Baseline vs 12 months smoking status.

**Figure 8 ijerph-18-03660-f008:**
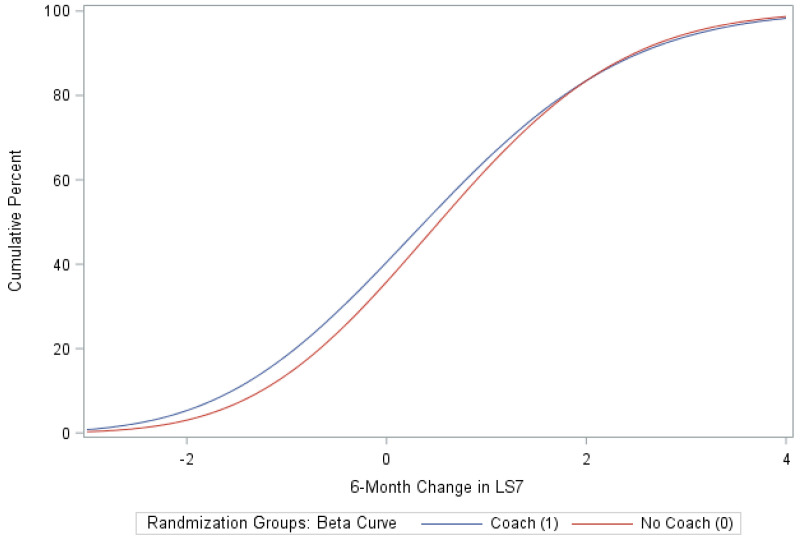
Cumulative distribution function of 6-month change in LS7 Fitted with a beta function.

**Table 1 ijerph-18-03660-t001:** Participants’ characteristics.

Baseline Characteristics	Treatment Assignment Coach vs. No Coach						
	All		Coach (*n* = 58)		No Coach (*n* = 62)		*p*-Value
	*n* (%)	Mean (SD)	*n* (%)	Mean (SD)	*n* (%)	Mean (SD)	
**Age at Baseline**	120 (100)	55.6 (8.87)	58 (100)	56.2 (8.32)	62 (100)	55.1 (9.09)	0.5112
**Sex at Birth**							
Females	80 (66.67)	-	40 (68.97)	-	40 (64.52)	-	0.6056
**Income**		-					
Less than $25,000	61 (50.83)	-	28 (48.28)	-	33 (53.23)	-	0.7993
$25,000 to <$50,000	26 (21.67)	-	16 (27.59)	-	10 (16.13)	-	0.3803
$50,000 or higher	28 (23.33)	-	12 (20.69)	-	16 (25.81)	-	Reference
Don’t Know/not sure	5 (4.17)	-	2 (3.45)	-	3 (4.84)	-	-
**Occupation**							
Employed	50 (41.67)	-	28 (48.28)	-	22 (35.48)	-	0.3572
Unemployed	31 (25.83)	-	12 (20.69)	-	19 (30.65)	-	0.5323
Retired	39 (32.50)	-	18 (31.03)	-	21 (33.87)	-	Reference
**Relationship Status**							
Divorced/Widowed/Separated	58 (48.33)	-	29 (50.00)	-	29 (46.77)	-	0.5348
Married/member of unmarried couple	34 (28.33)	-	17 (29.31)	-	17 (27.42)	-	0.5752
Never Married	28 (23.33)	-	12 (20.69)	-	16 (25.81)	-	Reference
**Census Tract Type (At risk, Resilient or none)**							
At risk	22 (18.33)	-	9 (19.52)	-	13 (20.97)	-	0.3638
Resilient	19 (15.83)	-	8 (13.79)	-	11 (17.74)	-	0.4449
None	79 (65.83)	-	41 (70.69)	-	38 (61.29)	-	Reference
**Highest level of education**							
High School or less	36 (30.00)	-	15 (25.86)	-	21 (33.87)	-	0.3398
Some college or higher	84 (70.00)	-	43 (74.14)	-	41 (66.13)	-	Reference
**Engagement with Health360x**							
Number of successful logins	104 (86.67)	6.9 (10.45)	46 (79.31)	7.5 (8.23)	58 (93.54)	6.4 (11.98)	0.5013
Number of sessions (e.g., app usage occurring after >=5 min of no activity_	104 (86.67)	8.1 (11.55)	46 (79.31)	7.7 (5.23)	58 (93.54)	8.5 (14.80)	0.7054
Median session duration	104 (86.67)	3.6 (4.09)	46 (79.31)	3.3 (3.40)	58 (93.54)	3.8 (4.57)	0.5174

**Table 2 ijerph-18-03660-t002:** Baseline and 6-month changes in Life’s Simple 7 (LS7).

LS7 and Key Risk Factors at Baseline and 6 Months	Treatment Assignment Coach vs. No Coach						
	All		Coach (*n* = 58)		No Coach (*n* = 62)		*p*-Value
	*n* (%)	Mean (SD)	*n* (%)	Mean (SD)	*n* (%)	Mean (SD)	
**LS7**							
LS7 at Baseline	120 (100)	6.2 (1.44)	58 (100)	6.3 (1.44)	62 (100)	6.1 (1.44)	0.1475
LS7 at 6 months	120 (100)	6.7 (1.72)	58 (100)	6.8 (1.84)	62 (100)	6.7 (1.61)	0.8323
Difference in LS7 at 6 months	120 (100)	0.5 (1.52)	58 (100)	0.4 (1.60)	62 (100)	0.6 (1.45)	0.6331
**Blood Pressure (mmHg)**							
Average Diastolic BP at Baseline	120 (100)	81.1 (9.81)	58 (100)	81.4 (9.75)	62 (100)	80.8 (9.93)	0.7424
Average Diastolic BP at 6 months	120 (100)	81.16 (10.84)	58 (100)	81.4 (9.82)	62 (100)	81.7 (11.79)	0.8723
Difference in Diastolic BP at 6 months	120 (100)	0.5 (10.78)	58 (100)	0.0 (10.48)	62 (100)	0.9 (11.13)	0.6453
Average Systolic BP at Baseline	120 (100)	133.9 (16.27)	58 (100)	133 (16.88)	62 (100)	134.7 (15.77)	0.5526
Average Systolic BP at 6 months	120 (100)	131.5 (18.67)	58 (100)	129.7 (17.31)	62 (100)	133.1 (19.86)	0.3230
Difference in Systolic BP at 6 months	120 (100)	−2.4 (18.47)	58 (100)	20.13 (62)	62 (100)	−1.6 (16.91)	0.6276
**Cholesterol (mg/dL)**							
High cholesterol at Baseline	79 (65.83)	-	42 (72.41)	-	37 (59.68)	-	0.1434
**Blood Sugar (mg/dL)**							
Glucose at Baseline	120 (100)	124.9 (60.25)	58 (100)	127.4 (72.77)	62 (100)	122.5 (46.05)	0.6503
Glucose at 6 months	120 (100)	122.8 (65.37)	58 (100)	113.8 (48.82)	62 (100)	131.2 (77.2)	0.1575
Difference in Blood Glucose at 6 months	120 (100)	−2.0 (66.46)	58 (100)	−13.6 (61.31)	62 (100)	8.8 (69.71)	0.0880
**Physical Activity (min)**							
Time spent in moderate exercise at Baseline	120 (100)	140.4 (175.19)	58 (100)	142.8 (206.8)	62 (100)	138.1 (140.95)	0.8818
Time spent in moderate exercise at 6 months	120 (100)	271.4 (685.57)	58 (100)	349.9 (930.47)	62 (100)	197.9 (310.72)	0.2695
**BMI (kg/m^2^)**							
BMI at Baseline	120 (100)	35.4 (7.64)	58 (100)	34.6 (7.73)	62 (100)	36.1 (7.56)	0.2876
BMI at 6 months	120 (100)	35.2 (8.43)	58 (100)	34.0 (8.29)	62 (100)	36.3 (8.48)	0.4497
Difference in BMI at 6 months	120 (100)	−0.2 (29.92)	58 (100)	−0.6 (2.31)	62 (100)	0.2 (3.37)	0.1601

**Table 3 ijerph-18-03660-t003:** Generalized Linear Mixed Models (GLMM) multivariate adjusted analyses of continuous variables.

**(a)**						
**Variables**	**Outcome: LS7 as a Continuous Variable at 6 Months**					
	**Treatment Assignment Coach vs. No Coach**	**Sex Male (1) Female (0)**	**Census Tract at Risk (1) Resilient (2) None of the Above (3)**	**Estimate**	**Standard Error**	***p*-Value**
Treatment Assignment	Coaching			−0.5210	0.3631	0.1548
Sex		Female		−0.3498	0.2964	0.2411
Treatment × Area	Coaching		At risk (1)	1.1269	0.5498	0.0433
Median lapsed time between sessions				−0.00002	8.599 × 10^−6^	0.0357
**(b)**						
**Variables**	**Outcome: BMI as a Continuous Variable at 6 Months**					
	**Treatment Assignment Coach (1) No Coach (0**	**Sex Male (1) Female (0)**	**Census Tract at Risk (1) Resilient (2) None of the Above (3)**	**Estimate**	**Standard Error**	***p*-Value**
Treatment Assignment	Coaching			−0.1896	0.6124	0.7576
Sex		Female		−1.6234	0.6299	0.0116
LS7				0.1399	0.2228	0.5315
Median Lapsed Time Between Sessions				0.00034	0.000019	0.0729
**(c)**						
**Variables**	**Outcome: DBP at 6 Months**					
	**Treatment Assignment Coach vs. No Coach**	**Sex Male (1) Female (0)**	**Census Tract at Risk (1) Resilient (2) None of the Above (3)**	**Estimate**	**Standard Error**	***p*-Value**
Treatment Assignment	Coaching (1)			0.7444	1.885	0.6944
Sex		Female (0)		−4.0721	1.9622	0.0409
Glucose				−0.01825	0.01536	0.2379
Area			Resilient (2) vs At Risk	15.4863	8.4346	0.0697
			None of the above vs At Risk	26.6194	16.3022	0.1061
Median Lapsed Time Between Sessions				0.000109	0.000057	0.0603

**Table 4 ijerph-18-03660-t004:** GLMM multivariate adjusted analyses of binary outcomes.

Binary Primary Outcomes	Odds Ratio	95% Cis		*p*-Value
Coach vs. No Coach groups comparison in Improvement (yes/no) in BP at 6 months	1.834	0.701	4.798	0.2135
Females vs. Males comparison of Improvement (yes/no) in BP at 6 months	2.394	0.853	6.716	0.0963
Coach vs. No Coach groups comparison in Improvement (yes/no) in LS7 at 6 months	1.047	0.411	2.667	0.9226
Females vs. Males comparison of Improvement (yes/no) in LS7 at 6 months	0.574	0.214	1.542	0.2676
Coach vs. No Coach groups comparison in Improvement (yes/no) in BG at 6 months	1.078	0.417	2.787	0.8754
Females vs. Males comparison of Improvement (yes/no) in BG at 6 months	0.457	0.157	1.330	0.1487
Coach vs. No Coach groups comparison in Improvement (yes/no) in BMI at 6 months	0.482	0.195	1.188	0.1113
Females vs. Males comparison of Improvement (yes/no) in BMI at 6 months	2.119	0.822	5.466	0.1187

## Data Availability

Data and code to generate results will be shared by authors upon request. Please contact the corresponding author.

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
