# Peer review of "Impact of Technology-Based Intervention for Improving Self-Management Behaviors in Black Adults with Poor Cardiovascular Health: A Randomized Control Trial"

_ijerph, 2021, doi:10.3390/ijerph18073660_

Round 1
Reviewer 1 Report
Thank you for the opportunity to read this manuscript (Manuscript ID: ijerph-1147526). I have some comments and questions which I will present below.
MAIN COMMENTS
INTRO section.
In the fourth paragraph of this section, the aim of the work and the research hypothesis are given. The aim of the study was to evaluate changes in health behaviors, and in my opinion it has been achieved. However, the research hypothesis assumes an improvement in health and I believe that its verification in this study is not possible. I raise such a doubt because it was not given (defined) what is deemed an improvement in individual health status, allowing for the fact that it would be bound to vary for different outcome variables (e.g. blood pressure, lipids, glycemia). For example it is widely accepted that 5% - 10% of weight loss, SBP/DBP reduction by 5 mm/Hg, FBG reduction by 20 mg/dl, HDL-C increase by 5 mg/dl, LDL-C reduction by 10 mg/dl, and TG reduction by 40 mg/dl are beneficial to one's health. In contrast, this study focused mainly on showing statistical (non-clinical) differences within and between groups. This only proves that the H0 assuming no difference was rejected with the simultaneous control type I error = 5%. The existence of such differences does not indicate that there is an improvement in health. Any improvement in the parameter value is not a significant improvement in terms of health.
METHODS section.
In this section, a lot of attention is paid to the description of the technologies used. That is good. But why was the information on laboratory testing of blood samples completely omitted? What methods were used? What are the reference ranges for each parameter? How was the information on the LDL-C value obtained? From the lab test or from the Friedewald's equation? I am asking this because the results of these tests were used to calculate the LS7. This issue also applies to BMI. Maybe it's obvious, but on the basis of what formula was the value of this indicator calculated? Paradoxically, the measurements of waist and hip circumferences are described in detail. There is also a lack of precise characterization of how much the participants in the study were burdened with risk factors for CVD.
ADDITIONAL COMMENTS
Is the table on page 7 of the manuscript a continuation of Table 1 or a separate table? Why are the descriptions of the results below the tables? Usually the opposite is true. Each table should be an independent entity. The fact that some information is given in the METHOD section does not mean that it cannot be repeated under the table in the form of a notes (e.g. what variables were basic adjustment, etc.). I get lost in the interpretation of the P value shown in Table 1. Sometimes it is the result of a comparison between groups, other times within a reference category. Can it be simplified somehow? Why are the measurement units of the presented variables omitted in Table 1? It looks very strange. It is not a good practice to repeat exactly the results in the tables in the text. I advise against. Figures 1-4 duplicate the results shown in the tables. Observing the mean values of the presented differences and the corresponding SD values in Table 1, the presence of large or very large variance is noticeable. SD is far from the mean. This means that the improvement of the tested parameters was varied. In my opinion, plotting the empirical cumulative distribution function for the observed improvements and estimating any percentiles would much better present the existing changes in parameters. As a consequence, it would also be possible to determine who obtained the greatest benefit from the intervention applied. And finally. The INTRO section is too large, the DISCUSSION section too shy.
In summary.
It is not my goal to demolish this manuscript. On the contrary, I really like the concept of this study. However, based on the methodology used, the Authors cannot talk about improving health, but about changing health behaviors. And a bit playfully. Please do not be surprised that the physical activity of women has decreased. They got access to more social media and they ran out of time to walk.
Author Response
Thank you very much for your feedback. Please see our responses attached.

Reviewer 2 Report
In this study, the authors demonstrate the benefit of behavioral intervention technologies among randomized Blacks with poor cardiovascular health. The results show that more than half of all participants experienced an improvement in Life's Simple 7, BMI, blood glucose, and SBP. Moreover, females demonstrated a significant improvement in BMI and diastolic blood pressure and a reduction in self-reported time spent in physical activities. However, no significant differences in changes expressed in LS7 between the technologies alone group versus the technology coupled with a health coach group. The study has been done meticulously and delivers clean and conclusive results. However, there are a few comments and concerns the authors need to address in this study:
- Since most of the results are represented as tables and bar graphs, for easy understanding the authors are requested to summarize the study by making a flowchart/illustration.
- The authors are requested to put p values in the bar graphs, mention the n number and denote which stat has been used to calculate the significance in the figure legends.
- The authors are requested to use abbreviation/nomenclature in the manuscript correctly. Eg. In the abstract, SBP has been used without prior describing the abbreviation. Again, when the authors describe Life's Simple 7 put LS7 in the brackets ‘Life's Simple 7 (LS7)’ so that when you use abbreviation next time you don’t have to explain again.
- The manuscript needs to check for syntax error/nomenclature/language thoroughly before getting published.
Author Response

(The authors gave the same response as above.)
